# How Gender Is Recognised in Economic and Education Policy Programmes and Initiatives: An Analysis of Nigerian State Policy Discourse

Ethel Ewoh-Odoyi 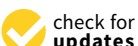

Turku School of Economics, University of Turku, 20500 Turku, Finland; erewod@utu.fi

**Abstract:** Many African states are involved in the frontline discourse on the fight for gender equality through the adoption of public policies, aiming to improve the lives of women through social, economic, and political development. In Nigeria, despite the adoption of Article 42 of the Constitution of the Federal Republic of Nigeria 1999 adapted from the United Nations principles of gender equality, which provides for equality and elimination of all forms of discrimination against women, the Nigerian state still struggles with different forms of gendered marginalisation issues against women in various aspects of Nigerian society; these issues are mainly due to cultural, economic, and legislative challenges. Therefore, this article explores how gender is recognized through public policy programmes and initiatives using a qualitative content analysis of relevant policy documents. The documents were collected from various government ministries and cover policy areas that represent entrepreneurship and economic activities in Nigeria between 2000 and 2020. The analysis confirms the recognition of gender in public policies by subjective bias and mediating access to education for female gender advancement in Nigerian society. Some gender gaps were also recognized and discussed in the article.

**Keywords:** gendered recognition; public policy programs and initiatives; qualitative content analysis; policy documents; Nigeria

## 1. Introduction

In recent years, there has been continued attention to public policies as an instrument for national development and poverty reduction. In developing countries worldwide, the dominant approach to attain this has been through the introduction of policies, programmes, measures, and activities for growth and economic development (Dao 2017). Policy programmes refer to policy objectives used by governments across the world to promote a wide range of activities. This includes, but is not limited to, educational policy, health policy, entrepreneurship policy, employment policy, agricultural policy, gender policy, housing policy, infrastructure development, and taxation policies.

Therefore, public policies do not only offer legal protection for the state in relation to rules, laws, and regulations, but through policies, programmes, and initiatives, governments provide support to address issues of socio-economic problems such as lack of finance, education, unemployment, cultural, religious, and gender discrimination (Ileana and Cornel 2017; Damon et al. 2016; Profeta 2020). While public policies, programs, and measures have been and continue to be implemented across the world and in developing countries, relatively little empirical research has been conducted into how gender is diffused through governmental public policies (Ahl and Nelson 2014). While existing research on women entrepreneurship and policy documents research has been focused on global north, more research is needed to delve into the context that investigates multifaceted societal gendered mechanisms (Henry et al. 2017). In view of the above-mentioned issues, the aim of this article is to analyse how gender is recognized

through public policy programmes and initiatives in the Nigerian policy context aimed at promoting or supporting female entrepreneurs.

In developing nations, policy measures aiming to improve the conditions of women through access to education and health have a limited impact due to constraints such as laws, norms, traditions, and codes of conducts through social institutions that hinder women (Morrisson and Jütting 2005). In Sub-Saharan Africa, especially in Nigeria, women are constrained by various social, economic, cultural, and political factors. As a result, the Nigerian governments have created and adopted several measures, policies, and programmes in support of women and other marginalized groups to promote entrepreneurship and economic activities such as the Youwin program, Sure-p, and Trader money programmes. However, scholars have reported that these policy programmes supporting entrepreneurship and women have been described to be inadequate or failed (Bolaji 2014; Bolaji et al. 2015; Edoho 2015; Drine and Grach 2012), as they have not provided the needed support to entrepreneurs and women. Furthermore, these policy programmes have not yet yielded much positive impact on women's enterprise development (Okeke-Uzodike et al. 2018) as such a proper policy coordination is needed for policy impact and stability (Afolabi 2015; Tende 2014).

In Sub-Saharan Africa, gender-blind policy measures are reported against women business owners (Vossenberg 2013), and the gender gap has continued to persist against women entrepreneurs in the economies of the developing countries (Vossenberg 2013). Scholars believes that policy makers need to be knowledgeable about policy programmes to ensure the effective use of public policy initiatives for the advancement of women business owners in Africa (Okeke-Uzodike et al. 2018). Others have argued that including gendered and feminist perspectives into all policies and programmes will benefit women (United Nations 1995, 2018; Orloff and Palier 2009) and that feminist theoretical approach to entrepreneurship promotion policy will become a fruitful and powerful premise for policymaking and policy program development to emerge (Vossenberg 2014) because unbiased policy programmes and initiatives will benefit men and women, as well as promote policy change.

The Nigerian nation has continued to adopt policies and programmes to improve the lives of women through various measures and activities with a focus on socio-economic and political development. However, even with the adoption of Article 42 of the Constitution of the Federal Republic of Nigeria (1999) that was adapted from the United Nations' principles of gender equality, which provides for equality and the elimination of all forms of discrimination against women, the Nigerian state still struggles with different forms of gendered marginalisation issues against women in various aspect of the Nigerian society in business (Mordi et al. 2010), in economy, and in politics (Eniola 2018). In March 2016, the Nigerian senate failed to enact the bill of "gender and equal opportunity", which forbids the physical, psychological, sexual, verbal, economic, social, and cultural abuse or similar mistreatment or mishandling which interferes with the integrity of a female or male human being (Makinde et al. 2017). These social, cultural, and legislative developments reflect the ways in which gender is acquiring visibility in contemporary Nigerian society and signify that gender recognition is an important and suitable area of research study.

Nevertheless, there has been a growing body of studies that recognises the importance and the needs for better policy discourse on the issues of gender for national development in Nigeria (see Para-Mallam 2007; Soetan and Akanji 2019). However, these studies have investigated policy issues through women's underrepresentation in academics, in politics, and marginalisation in economy, education, and the rights of women across Nigeria (see Para-Mallam 2007, 2010; Soetan and Akanji 2019; Aderemi 2019; Muoghalu and Eboiyehi 2018; Idike et al. 2020; Bako and Syed 2018; Ekhator 2018). These issues have persisted against women's contribution to Nigeria's national development. However, these available studies have not explored the issues of gender recognition in the Nigerian public policy context of documents analysis. In view of the above-mentioned issues, this article does not question nor analyse how public policy benefits and works for

women. Instead, the aim is to explore how gender is recognised through public policy programmes and initiatives in the Nigerian policy context.

This article uses a qualitative content analytical approach to analyse policy programmes supporting entrepreneurship and economic activities in Nigeria. This article answers the question of "How do public policies recognize gender in policy programmes and initiatives" in a Nigerian policy context. Most studies on women's entrepreneurship policy and document analyses have been based on the Global North (Ahl and Nelson 2014; Henry et al. 2017), that is, countries that share similar economies such as the US, Canada, Western Europe, and Northern Europe. These research results echo contexts and policies suitable for the Global North but not for developing economies of the Global South, such as Sub-Saharan countries.

This article can be used to inform practice, as policy makers can use the findings of this article to better understand the issue of gender in policy programmes. The findings can be used by policy makers and researcher to better access the needs of women in every entrepreneurial environment to create appropriate support policies for women. By doing so, it will improve the environment for developing women in general. This article contributes to several scholarly discussions. As noted by (Naude 2010, 2013), designing and introducing policies to promote entrepreneurship in developing countries seems to become challenging and to be more complex in the future. This is also the case in developing countries such as Nigeria, where public policies seldomly struggle to meet the required level of equality.

First, the article contributes to the discussion of gender and public policy literature, especially on the improvement and impact of governmental policy initiatives on the development of women entrepreneurs in both developed and developing economies (Ahl and Nelson 2014; Henry et al. 2017; Panda 2018).

Thus, the recognition of gender in policy programmes exposes the weaknesses of a gender-biased policy against women, in relation to the modern public policy standard that should not discriminate against women's position in society. Second, the article contributes to the discussion of public policy programme implementation from a gendered perspective. Thirdly, this article has revealed how women are marginalised through public policy and suggests women's education for gender advancement in Nigeria. It also contributes to the debates on issues of gender marginalisation through public policies and how institutions can be strengthened for the benefit of women in developing countries (Minniti and Naudé 2010). The article also contributes to the issues of gender, and women's changing position in African societies (Medie 2019). Finally, this article contributes to the discussion of practical issues of women's access to education for gender advancement in the Global South and Sub-Saharan Africa using a Nigerian example.

This article focuses on policy programmes supporting entrepreneurship and economic activities on how gender is recognised through public policy programmes. To address these issues, this article uses a historical perspective to examine the norms of public policy in relation to gendered recognition of women in policy programmes and initiatives within a 20-year period from 2000–2020 through a qualitative content analysis of Nigerian policy documents. The article is structured as follows: Section 1 is an overview on policy issues on gender. Section 2 describes the research context on the treatment of gender. Section 3 describes the methodology analysis, the research material, and the data used in the study. Section 4 describes the findings, and Section 5 provides the discussion and conclusion.

*Analysing Public Policy and Gender*

The term "policy" represents a diversity of meanings to scholars (Arshed et al. 2018). A policy can be defined as a plan of action agreed and chosen by a group of people, organisation, or political party (Akinyemi and Adejumo 2018) and the action can guide a wide range of related activities in various spheres (Mackay and Shaxton 2011), and the activities can be in the form of institutional structures for guiding the common activities of the participating actors (Arshed et al. 2014). The process of policy is typically seen as

having a series of sequential parts, such as problem occurrence, agenda-setting, options for consideration, policy formulation, adoption, implementation, and the process of evaluation. Public policies are often used by governments to solve "wicked problems" that are resistant to change (Hudson et al. 2019), or to promote entrepreneurship, social inclusion and social change, ethics, gender equality, economic growth and development Acs and Szerb (2007), as well as prompting of innovations (Hyytinen and Toivanen 2005; Basant 2018). Governments across the world use public policies to provide regulatory frameworks for every citizen, for example, macro-level policies can be used to solve issues socioeconomic issues such as poverty, unemployment, and economic growth (Mckeown 2016). This study aims to explore how public policies recognise gender in policy programmes and initiatives.

In entrepreneurship literature, the issue of gender in public policy has been described as presenting a challenge for policy initiatives across the world (OECD/European Union 2017; ILO 2012). Studies on public policies have investigated start-ups, economic growth, performance (Acs and Szerb 2007), and the promotion of entrepreneurial activities in different economic context. However, research focusing on women in the policy arena highlights how gender issues have been a focus of public policies. For example, scholars believed that various structures of policy practices have becomes obstacles to the advancement of gender-equality through policy-making mechanisms (Schofield and Goodwin 2005). As the role of government and public policy issues in entrepreneurship literature is seemingly an understudied research area, with only about 4% of research conducted empirically and theoretically (Link et al. 2016), this has highlighted public policy and gender as an important area of study.

There is a growing recognition among researchers and policy makers that there are extensive and deep implications as regards gender-based discriminations through policy programmes and initiatives (UNDP 2013). There has been criticism about the gendered effect of public policies (Ahl and Nelson 2014) that suggest possible obstacle for gender equality (Schofield and Goodwin 2005). As access to mainstreaming policy support is implicitly gender-biased (Welter 2004), therefore, most national policy mechanisms that promote gender equality have not represented women business owners well in most government policy initiatives but believe that feminist perspectives in policy development will only benefit women (Pettersson et al. 2017).

## 2. The Nigerian Context for Gender in Policies and in Economy

The history of gender in Nigeria can be dated back through three specific eras: the pre-colonial era, the colonial era, and the post-colonial era (Abdul et al. 2011). During the pre-colonial era, Nigeria's political structure was purely monarchical. The nation was made up of diverse societies and kingdoms of which women participated actively in both the private and public spheres and influenced the socio-political scenes of the various regions (Eniola 2018). However, after the British amalgamation of the northern and southern protectorate, a colonial type of administration was formed in 1914. During this time, patrilineal and patriarchal kinship structures dominated Nigerian societies, which brought about the ideology that women should stay at home and take care of the family; during this time, they were expected to engage in work considered complementary to men (Aderemi 2019, p. 84). These principles made a clear distinction between men and women both culturally and religiously in Nigerian society.

However, independence in 1960 ushered in a parliamentary system of government that lacked emphasis on gender. During this time, the national government dictated the lifestyles of women by endorsing the domesticity of women and the unwaged services provided by women to the family. Much of the legislation attempted to control women's sexuality, as well as fertility through defining their subordination (Aderemi 2019, p. 83). The division of labour along gender lines was more pronounced in the country's industries, as women in rural areas farmed and sold homemade products to support their families. Men in the south cultivated yams and women cultivate beans and cassava (Aderemi 2019, p. 84).

Despite these segregations across gender lines, providing education for young people was a priority, but there were huge disparities throughout the regions in term of women's economic development. For example, during the post-colonial period, there were more girls who were not in school in the North, where poverty was prevalent compared to the oil-rich South (Aderemi 2019, p. 83). This was due to the gender-biased economic policies of the 1980s, which pursued economic growth through structural adjustment and extensive liberalisation that led to what is today referred to as the 'feminisation of poverty', with Nigerian women bearing the effect of the economic melt-down which was aggravated by high rates of unemployment and the loss of jobs for many Nigerians families (Boyi 2019). In addition, there were differences in regional disparities in terms of culture and religious practices across Nigeria. In the North, there was a low-level of participation by women in entrepreneurship, politics, and public life compared to the South. Women in Southern Nigeria gained the right to vote in 1959, but it was not until 1979 when women in the North could vote (Omoluabi et al. 2014).

However, public policy programmes supporting entrepreneurship and economic activities in Nigeria date back to the colonial era. After Nigerian independence from Britain in 1960, Nigeria formed a civilian government, which was then overthrown by military coup d'état in 1966. The military-led government lasted from 1966 to 1999; this era was characterised by instability, coups, and countercoups. The long stay of the military in the politics and leadership of Nigeria has been characterised by changing socioeconomic structures and policy institutions (Soetan and Akanji 2019, p. 1), which has led to the possible underdevelopment of Nigeria between 1966 and 1999 (Idike et al. 2020).

In 1999, Nigeria ushered in a democratic system of government, leading to a more realistic standard according to the world's assessment of democracy. The government at that time was led by Major General Olusegun Obasanjo, since then, and until the current time, Nigeria has experienced stability in its governance. In 2004, fiscal policies such as NEEDS, SEEDS, and LEEDS were enacted to solve issues of socioeconomics and sociocultural well-being among men and women across Nigeria. These programmes were targeted at sustainable growth and poverty reduction, with the aim of empowering people and improving social service delivery, fostering economic growth in non-oil sectors, and enhancing effectiveness and efficiency in governance (Central Bank of Nigeria 2005). The rationale was due to an increased percentage of Nigerian living below the poverty line between 1980 and 1996 (The World Bank 1996) with expenditures of less than two-thirds of the average per capita household expenditures, an increase from 28 percent to 43 percent (Canagarajan et al. 1997). The preamble to the fiscal policies such as the NEEDS strategy was to put all gendered issues into perspective by putting them into policies, thus aiding a national transformation towards poverty reduction and equitable growth.

Furthermore, between the years 2000 and 2006 a new national gender policy was adopted and implemented to address the issues of gender inequality across the country. Policies aimed at addressing the economics of gender in development has since been programmed through different national public policies programmes and initiatives. Since then, public policy initiatives, from one political regime to another, have been aimed at promoting gender equality through entrepreneurship and economic activities. The adoption was made possible due to the uncoordinated national response to women's question in Nigeria that was required by the United Nations Sustainable Development Goals in underlining macroeconomic progress in all population and the rational to see how the Nigerian state has made progress in the absence of multidimensional poverty, women's empowerment, and gender equality (Soetan and Akanji 2019, p. 1).

Presently, gender is seen as a priority and a key tool for national development in Nigeria. The question is, how is gender to be recognised in policy programmes and initiatives? The section 42 of the Nigerian Constitution provides for the equality and elimination of all forms of discrimination against men and women as required by the United Nations. Despite the national average of female secondary school enrolment increasing from 45.3% in 2010 to 65.8% in 2018, discrimination persists in the national

and state status of women. Marginalisation against women in various aspects of Nigerian society remains prevalent, particularly in the cultural, social, political, and economic decision making, thus, it has remained a challenge for gender equality.

Culturally, women in Nigeria experience a variety of cultural issues, such as child marriages, which are one of the major obstacles for women's advancement. Female genital mutilation has remained a widely prevalent phenomenon in Nigeria (Yaya and Ghose 2018), with an increasing number of women experiencing this practice with a national prevalence of 41% (Okeke et al. 2012). FGM is detrimental to women's health (Nour 2015) and thought to negatively affect women's education and other opportunities for development (Rogo et al. 2007).

In Nigerian society, gender roles are also manifested in social rights and entitlements in a form that denies women equal economic and political opportunities, especially the right of women to the ownership of land (Ajala 2017). The dispossession of women's right to inheritance, or ownership of land and other properties is one of the major barriers to the recognition of women's economic rights in Africa (Techane 2017) as lack of access to land, as a means of economic productivity can create women's vulnerability in overcoming poverty and violate their human rights. In many African societies, the issues of landed property ownership are often transferred within the family due to customary laws that often exclude women and girls from ownership and inheritance (Techane 2017; Efobi et al. 2019). In addition, traditional patrilineal advantages in Nigeria give preference to male children over inheritance (Nwokocha 2007).

In politics, the national average of women's political representation in Nigeria has remained at 6.7% in elective and appointive political offices (Idike et al. 2020), undoubtedly below the international average of 22.5%, the African regional figure of 23.4%, and the West African sub-regional figure of 15% (Oloyede 2016). There are also no female members in the Nigerian Armed Forces Ruling Council, which implies that women play no significant role in the decision of the central government (Oloyede 2016). Of the 84 million registered voters in the 2019 election in Nigeria, women account for almost 40 million, 47.14%, yet there are few women representatives in Nigeria's national politics (Onyeji 2019), which has resulted in little female influence on policy decisions. These issues show how women are treated in Nigerian society.

## 3. Methodology: Research Design, Materials, and Analysis

The research data for this study consist of 17 national policy documents that are related to policies supporting entrepreneurship and economic activities in Nigeria between 2000 and 2020. The rationale for choosing these documents was because they were the available data that are suitable for analysis on gender recognition in policy documents. The policy documents were published and gathered mostly from e-portals of the Nigerian investment promotion commission (www.nipc.gov.ng, accessed on 30 December 2019), and from various federal government ministry portals. These portals are mainly used by the government for disseminating public information and making policies available online. The criteria for the data collection were first that the policy documents had to be published by the government on issues relating to public policy actions on entrepreneurship and economic activities. The data gathering took place between October and December 2019 (see Appendix A Figure A1 for the data gathering, processing, and analysis flowchart). Although there are a variety of materials that can be used for gender recognition policy, as policy documents may not be the only best option for data, as such, the flows in data may not be perfect as they are the only ones available online. The data were collected to cover policy actions of the governments from various ministries, see the Appendix A for document material used in the study (see Table A1 in Appendix A).

The next step followed a data screening process to check if the collected research documents could answer the research question "How is gender is recognised in policy documents?" The screening of the documents resulted in a total number of 13 national policy documents for analysis with a primary focus on documents relating to governmental public

policy programmes between 2000 and 2020. The rational for choosing this time-period was a result of the previous military administrations, which ended in 1999. Since the year 2000, Nigeria has had a democratic system of government without a coup d'état, and this has brought about stability in government actions. The screening excluded four national policy documents, these concerned industrial policy, taxation, national petroleum policy, and National Gender policy. These documents were excluded because they did not answer the research question. The selected papers were chosen to answer the following research question: "How is gender recognised in policy text" (Eriksson and Kovalainen 2016).

*Data Analysis*

The analysis focused on how gender is recognised through public policy programmes and initiatives. The analysis is based on the social constructionist and inductive approaches. The data have been analysed using both qualitative content and the thematic analysis of the research content and the themes that emerged from the research materials. This process was aimed at providing a complete and accurate description of the studied phenomena (Eriksson and Kovalainen 2016). The data analysis of the text involved a wide range of analytical techniques (Saldaña 2013; Eriksson and Kovalainen 2016) such as reading, coding, sorting, and categorisation of the data.

Firstly, the data were read to understand if the content of the data answered the research question, then followed by a re-reading of the data again with each description of the documents by selecting sentences such as words and phrases that addressed the research question. The second step followed using NVivo coding to reduce the large quantities of the data into a form of data that could be easily handled and to understand the significance within the textual data by displaying in NVivo cloud the clusters with the most frequencies between the data text. This process focused on understanding the relationship between key themes and patterns differences in the text by mapping the most important themes in understanding the textual data. Since NVivo cannot interpret the whole data, the next step was a descriptive manual coding of those sentences and phrases in line with the research questions in the Microsoft Word documents. This had the aim of identifying the stated motivation for the narratives.

The third step was a sorting of the data by grouping the themes into different categories, followed by an understanding of the most frequently used words within the data and then connecting the emerged themes based on their categorisation into a meaningful construction of how gender is recognised (See Table A2 and Figure A2 word cloud of the most frequent words in policy documents). These themes were then analysed using a thematic analysis of what has now shown to provides the empirical basis for the evaluation of the key theoretical claims concerning public policy recognition of gender in the Nigerian context.

The fourth step was an exploration of the themes that arose from the text data: The six most interesting themes that appeared in the document material of the analysis. The themes were education, gender, access, discrimination, implementation, and cultural factors. Among these six themes there were two top principal themes that emerged (education and gender). The emergence of these themes was not a surprise, as other literature has emphasized the issue of gender access to education (Para-Mallam 2010), and this research does not represent a non-empirical or self-evident truth concerning gender recognition in Nigerian policy documents. Interestingly, access, discrimination, implementation, and cultural factors were also very much emphasized in the policy documents.

The next-level of analysis was based on how the first-order categories which were connected and grouped into themes that could provide more coherent categories that could describe "how gender is recognised" in the policy document. An abductive process was followed for interpretation within the textual data on how gender is recognised within public policy. The rationale for this level of analysis process was to determine how gender is recognised in public policy text. In exploring this classification level, two main themes were presented in the textual data to communicate the multifaceted connection between

the data. The analysis of this classification was followed by a list of thematic categories on the description of each meaning of the six themes presented in the data.

## 4. Results: Gendered Recognition in Policy Document

Generally, in the selected policy documents, gender was present. The exception was that the national gender policy paper did not answer the research question. However, there was an absence of gender in taxation policy, national petroleum policy, and industrial policy documents. In the analysis, there are more words on issues concerning gender description and on issues affecting females rather than males in the policy documents. The analysis of the selected policy documents shows a gender-biased policy by mediating "Education", which is access to basic primary and secondary education is a problem for female gender advancement.

### 4.1. Education

The national average of school children not in school in 2018 was 12.7 million, but presently, Nigeria has an increased rate of 14 million children not in school in 2020 (Ogunnaike 2020). As the gender gap widens, education is seen as a high priority area, with very large regional disparities across Nigeria. In the North, there is a very low school attendance rate of 35.6% of children in primary and early childhood education, therefore, only 53% of children have a clear daily attendance in school. Girls in the North have an attendance rate of 47.3–47.7%, which means that half of girls are not in school in the North. In the South, there are schoolboys and girls not in school, which has created the high number of about 60% children not in school across Nigeria, and this will contribute to a high 60% illiterate adult population in the future. The low attendance of school children in Nigeria has been characterised by various factors, such as poverty, economic demands, and socio-cultural factors that hinder girls' education in policy documents.

The main challenges facing the Nigerian educational system according to the policy documents are issues of; education quality, which is linked to the inequality of both rural and urban schools, poor education facilities linked to the lack of maintenance in school infrastructures due to inadequate funding, non-implementation of the national teachers' policy, the gender imbalance of educators, gender imbalance in the enrolment of girls, and the low enrolment of girls in STEM and TVET education. Gender educational imbalance in both primary and secondary education and the children not in school, of which 60% percent are girls, is due to poverty and sociocultural barriers that are against girl-child education in the policy documents. The gender imbalance of female and male teachers has contributed to lack of female role models in teaching programmes. Women also experience sexual harassment and other social vices in school, with unfriendly gender educational facilities and the distance from schools, which help to create gender gaps in education. Other issues such as limited job opportunities, child labour, early marriage, poverty, illiteracy of parents, over population, urbanisation, and economic poverty all obstruct girl-child education.

Presently, the enrolment of girls may have increased, but between the year 2000 and 2010, girls not in school were 42.6–45.7%, while enrolment was 44.7–45.7% for girls. Meanwhile, the proportion out of school primary boys was 28.1–28.2%, and enrolment was 55.3–57.6% for boys. In addition, girls' enrolment in secondary school increased from 45.3% in 2010 to about 45.7% in 2015, and in tertiary institutions, enrolment increased from 43.2–43.7% (National Bureau of Statistics (NBS) 2018). Although there has been a slight improvement in the educational enrolment of girls, there remains the challenge that the issue of access to education has been inadequately addressed. The national education policy was expected to put in place an egalitarian society through the educational system. This was to be achieved by addressing the issues of education access through mainstreaming gender within the educational sector with the focus placed on girl-child education and school enrolment. This expectation of egalitarian equality has since failed due to many issues affecting the educational sector.

## 4.2. Gender

In addressing the issues of gender in Nigerian policy documents, gender is described in ways that describe the activities of men, women, children, and youths in the treatment of gender. However, there are commonalities in the treatment of gender in the policy documents, e.g., the treatment of women in the use of access to opportunities, services, and resources that are beneficial in the participation and responsibilities in both the private and public spheres. "Gender" in the policy documents was linked to the issues of ad-dressing basic education imbalance through the promotion of the gender mainstreaming of girl child education. Gender advocacy concerned issues of contraceptives, child labour, and the promotion of gender-responsive programmes and activities in all policy areas. Gender in the policy documents is linked to the issues of education inequality and suggests possible gender advocacy in policy areas and at all levels of government to tackle the issue of gender in Nigerian society.

## 4.3. Access

Women experience difficulties gaining access to resources that will be beneficial to their development. For example, access to finance is a problem for female business owners, as banks require collateral to access financial credit. In addition, women have limited access to contraceptives due to a lack of institutional mechanisms to make them available for free. Other factors limiting access to contraceptives are related to cultural, social, and religious issues. For example, the permission of husbands to access services is essential; about 36.2% of women in Nigeria have an unmet need for access to contraceptives. The lack of access to education for young girls, especially in northern Nigeria, has been attributed to violence and terrorism. Most young girls have been kidnapped in schools across the North by the Boko Haram terrorist group (Amanda 2018), which has highly influenced the education of both boys and girls, with the latter bearing a tremendous burden. The persistent discrimination in access to resources in the Nigerian policy text revealed the need for gender mainstreaming, especially in teaching the consciousness and awareness of gender equality in Nigeria. However, the lack of political commitment to gender equality in government institutions responsible for promoting equality is generally weak with minimal resources. Since 2020, most ministries have allocated no budget for mainstreaming gender issues from annual national fiscal policies (Nelson 2020).

## 4.4. Discrimination

For example, issues of gender discrimination in the access of women to education, and discrimination in land ownership against women were reportedly due to socio-cultural barriers that limit women from land ownership. In addition, sexual discrimination in employment against women is related to gender in the workplace, these especially affect women as regards employment recruitment, promotions, training, inadequate provision of maternity leave, health-care protection, and equal remuneration. Women also experience marginalisation in access to the use of technology and information regarding agricultural outputs and issues concerning climate change.

## 4.5. Implementation

Public policy implementation is one of the significant issues facing Nigeria in its effort to achieve national development (Ahmed and Dantata 2016). Implementation in policy documents emphasizes the issues of the policy implementation gap. Poor policy implementation played a crucial role in re-enforcing gender-based discrimination in policy areas. For example, the weak implementation of gender-based policies, weak legal framework to protect and promote the welfare of women and children, and the lack of implementation principles in eliminating all forms of discrimination against women has been attributed to the structure of Nigerian legislation, which gives men more authority over the decision-making processes that affect women. There is a general lack of awareness and pervasive influence over issues concerning the importance of gender equality among

policymakers. Policymakers are unable to manage their differences due to ethnic, cultural, and religious issues on national issues that affect their different groups due to institutional biases. Additionally, the inconsistent and ineffectiveness of gender-responsive programmes and uncoordinated policy responses may be attributed to changes in governance, policy principles, corruption, and the low budget allocation of funds to solve policy issues.

*4.6. Cultural*

In the policy documents, cultural factors are a threat to women's recognition in Nigerian society. The issues of harmful traditional and religious practices against women are linked to the culture that reinforces gender roles and lifestyles of men and women in Nigeria. Sociocultural barriers in the policy text are also connected to socioeconomic issues. For example, cultural barriers impede female participation in basic education, which also contributes to early marriages among young girls. These issues are mostly prevalent among Northern Nigerians, where 68% of women between the ages of 20–49 were married before the age of eighteen (UNICEF 2017). Furthermore, there are sociocultural issues against women in access to basic education, land, and inheritance, these factors help to reinforce socioeconomic issues such as poverty, child labour, low level of education among girls. Also, these cultural and economic issues create legislative loopholes for an effectives policies and laws constrain women's progress in economic activities and development.

**5. Discussion and Conclusions**

This study analysed how gender is recognised in policy documents to demonstrate an actual policy setting on how gender issues are integrated into public policies for the benefits of equality. In the analysis of the policy documents, this article has examined how gender is recognised through Nigerian public policy documents in the fields of Education, Health, Agriculture, Social protection, Entrepreneurship and Information and Communications Technology, Information and Communications Technology in education, Employment, Child labour, Micro, Small, and Medium Enterprises, Food and nutrition, Infant and young child feeding, and Family planning policies. The analysis shows that gender is recognised by subjective bias by mediating education. Furthermore, the word education has more themes arising from textual data, followed by gender, access, discrimination, implementation, and cultural factors. This analysis demonstrates that gender is recognised through policies with a specific focus on education and that mainstreaming gender issues is a way of combatting discrimination in access to resources. Therefore, cultural challenges can be solved primarily through policy implementation.

The data shows the general neglect of female education in creating equality. This interpretation means that improving female access to education can help enhance the status of women in Nigeria. The views of gender discrimination in accessing education are due to harmful gender stereotypes concerning the roles of women and men in Nigerian society. Discrimination in access to land ownership stands as a vital challenge to women's development, as land is seen as an asset in agricultural food production and shelter development. Gender discrimination in land ownership leads to the segregation of women in land ownership, which hinders women's access to land for agricultural purposes and for the sale of land to raise funds or as collateral security for business loan.

Institutional mechanisms contribute to discrimination in access to finance, adequate maternity and healthcare, and access to agricultural inputs and information. Access to business finance and other regulatory requirements in the business environment mandates the provision of collateral to banks to receive business financial support. Thus, women owners are often perceived as high risk by lending institutions due to their inability to provide needed collateral. In addition, the inadequate maternity and healthcare protection against women is a challenge as the Nigerian Labour Act provides only 12 weeks of maternity leave for pregnant women. During this period, the employee concerned shall be eligible for at least 50% of her wages. Furthermore, the discrimination of women in employment recruitments, promotions, training, and equal remuneration links with gender

discrimination in the workplace policy context, leading to women's marginalisation in the workplace. In addition, the lack of access to agricultural inputs and information regarding the use of agricultural technology and lack of access to general basic institutional support is a challenge for equality.

However, in the textual data, cultural factors reinforced discrimination against women in access to resources in various context. For example, cultural and religious issues relating to women's access to education, land, contraceptives, and other harmful traditional practices against women puts them in a subordinate position that affects women's development in Nigeria. The cultural themes in the data reinforce discrimination by creating social, cultural, and economic issues for women. For example, the cultural barriers that impede female participation in basic education and their access to land and contraceptives, impact on gender equality and create poverty, leading to many social and economic problems among women in Nigeria.

In Nigeria, women face a considerable number of impediments due to deeply rooted discriminatory via sociocultural, traditional value that are embedded in policies, legal environments, and in the institutional support environment. The cultural issues are often rooted in the family, religion, history, ideology, and culture of the people, it is not surprising that cultural ideologies about how women are treated appear to be common in the policy text and shows up a lot in the discriminatory narratives. Previous studies have also pointed out the issues of women's marginalisation (Bako and Syed 2018) and the lack of the implementation of anti-discriminatory laws to the discrimination of women in Nigeria (Chegwe 2014).

Scholars believe that the ideology of gender is the reason for gender discrimination, given that gender ideology creates differences and similarities on a common understanding of men and women (Philips 2001) in relating to society. In Nigeria, cultural and religious ideologies contribute to the discrimination of women. Nigerian society differentiates between equality laws and cultural practices, as cultural and religious ideologies appear eminent in the treatment of gender issues. The Nigerian national laws have emanated from several different ideologies such as the common law, a customary law which also integrates religious such as Islamic and Christianity values (Chegwe 2014). As a result, these laws are exempt on the principles of equality and have created issues of marginalization that have affected women in different spheres of Nigerian society.

First, this article draws from the discussion of gender in public policy literature from the Global North, as most studies on women's entrepreneurship and policy document analyses have been based on the Global North (Ahl and Nelson 2014; Henry et al. 2017). The research results of these studies echo contexts and policies suitable for the Global North but not the Global South, such as sub-Saharan countries.

From the analysis of the policy documents, gender has been integrated into Nigerian public policy since the year 2000, and this has followed the adoption the United Nations' development goals for gender equality and development. Since then, gender has been a focus of public policy, with a specific focus on female access to education. However, there has been a slight change in the increase in female enrolment in schools, as Nigeria has continuously continued to promote equality in all walks of life through various policies and programmes targeted at gender development. However, this has not yielded much impact on equality as there still exists a high prevalence of gender discrimination due to a lack of access in socioeconomic opportunities, cultural and religious factors, poverty, and inadequate legal and policy frameworks.

The issue may not be public policies per se, rather, the mechanism and inconsistencies in the policy process, practices, and the lack of policy implementation have adversely affected women's participation in all spheres of Nigerian society. As such, the outputs and outcome of policy implementation processes are often determined by a complex interaction of factors (Medie 2013), which requires the responsibility of all stakeholders in both government and NGOs for equality in all aspects of policy development to avoid the marginalization of gender to specific units or policies. Nigerian public policies contribute

to discrimination, and this is due to the embedded nature of the cultural and religious ideologies, intertwining with national laws and policies. The national laws such as customary and religious law collide with institutional policies, which may be an important factor for discrimination in policy and on how gender is recognized and treated in a Nigerian policy context. The suggestion of mainstreaming gender issues is important in addressing all forms of gender inequality in society, and it can be an opportunity to create the awareness needed in all areas of public policy in achieving economic development and gender equality in Nigeria.

This article explores how public policy recognised gender in policy programmes using policy documents as an analytical tool. The authors acknowledge the limitations of the study. Personal bias and preference may have unintentionally influenced the paper identification and selection of text documents and the policy programmes that were used for investigation. Subjective biases and validation may be inherent in qualitative content and thematic analysis, which may have created minor discrepancies in the interpretation of data. The implication of a bias in public policies may interrupt the plans of public policies in meeting the demands for gender equality. Having an equal gender policy can be dependent on several factors in different contexts of cultures, laws, and traditions, and in this research context, more efforts need to be made for women in achieving an effective public policy objective. For example, family and social protection policies need to meet the demands for gender equality.

The findings from this study propose several avenues for future research. First, there is a need for an understanding of how policies and programmes benefit and work for women and men. This will help in understanding the needs of men and women in relation to policy support programmes. Secondly, research is needed to further explore gendered influences of authorship of policy documents, specifically, to focus on issues of policy process, design, and implementation, that is, the roles and the different interests of actors involved in the policy process, specifically, how the policy resources are managed and whom they are made for in the long term. Scholars can also explore how feminist perspectives can influence better public policy promotion for the benefit of gender equality.

**Funding:** This research received external funding from Kone Foundation with the grant number 201901690.

**Institutional Review Board Statement:** Not applicable.

**Informed Consent Statement:** Not applicable.

**Data Availability Statement:** Data is contained within the article.

**Acknowledgments:** This work was supported by Turku University. Also, this research received external funding from Kone foundation Finland. Also, special thanks go to anonymous peer reviewers whose comments has been helpful.

**Conflicts of Interest:** The authors declare no conflict of interest.

# Appendix A

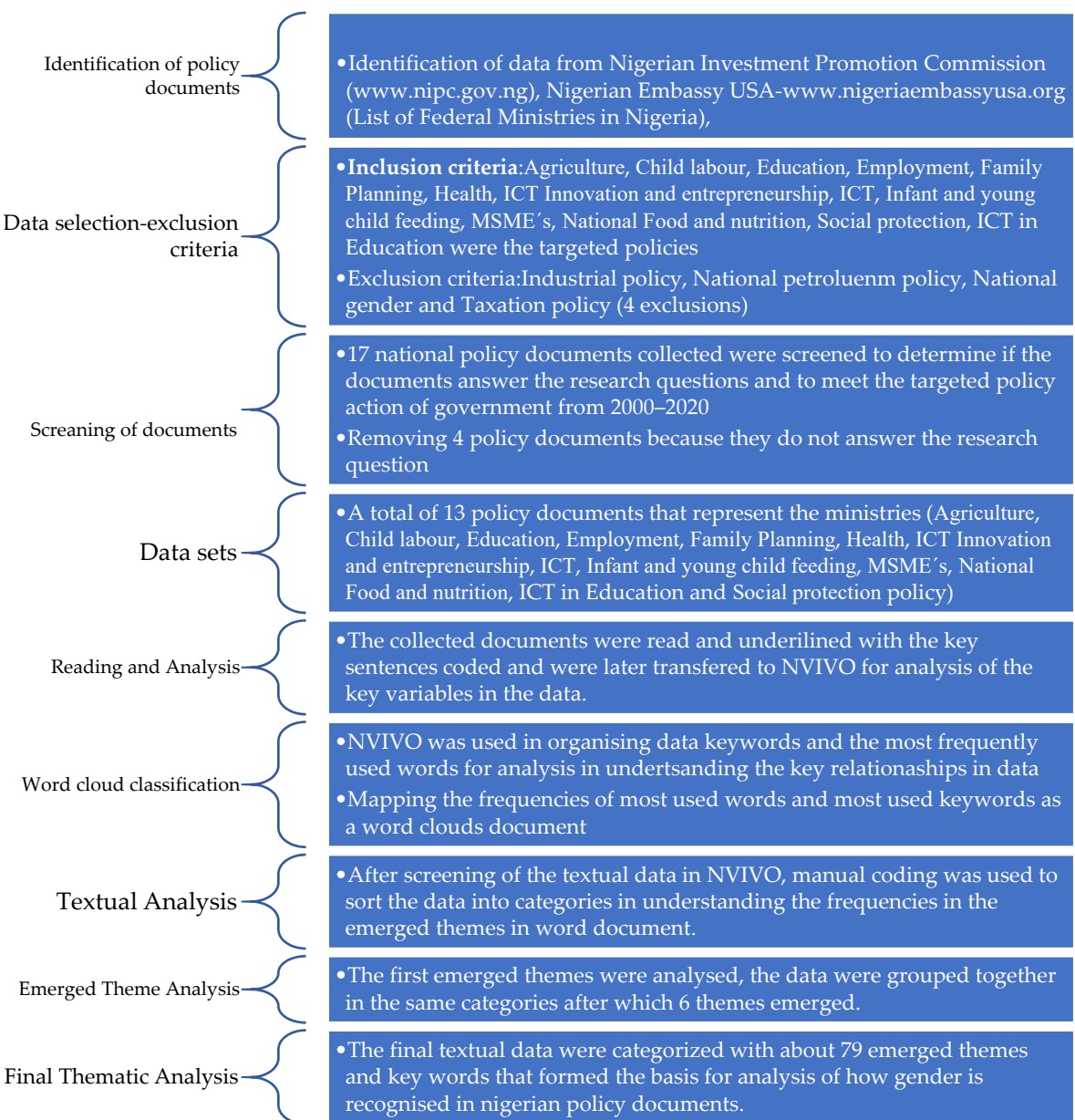

**Figure A1.** Data selection process, classification and analysis flow chart.

**Table A1.** Document materials used in the study.

|  | Documents on the Public Policy Programmes | Exclusion | Year |
|---|---|---|---|
| 1 | National food and nutrition policy |  | 2001 |
| 2 | ICT Innovation and entrepreneurship policy |  | 2001–2025 |
| 3 | Infant and young child feeding policy |  | 2008 |
| 4 | National Gender Policy | Excluded | 2008–2013 |
| 5 | Taxation Policy | Excluded | 2012 |
| 6 | ICT policy |  | 2012 |
| 7 | Child labour policy |  | 2013 |
| 8 | Nigeria Industrial Revolution Plan | Excluded | 2014 |
| 9 | Family Planning policy |  | 2014 |
| 10 | MSME's policy |  | 2015–2025 |

**Table A1.** *Cont.*

|  | Documents on the Public Policy Programmes | Exclusion | Year |
|---|---|---|---|
| 11 | Agriculture policy | | 2016–2020 |
| 12 | Health policy | | 2016–2020 |
| 13 | The National Petroleum Policy | Excluded | 2017 |
| 14 | Employment policy | | 2017 |
| 15 | Social protection policy | | 2017 |
| 16 | Educational policy | | 2018–2022 |
| 17 | ICT in Education policy | | 2019 |

**Table A2.** How Gender is recognised in the Nigerian public policy documents.

| Themes | Gendered Recognition Coded and Categorised Themes |
|---|---|
| Education (18) | 12.7 million out-of-school children<br>60% percent of girls are not in school<br>Females account for nearly 60 per cent of the country's illiterate population<br>Girls of school age constitute 60 per cent of population<br>High numbers of absenteeism/children not in school in North<br>Inadequate educational funding<br>Low Boy enrolment in STEM<br>Low female enrolment and retention in ST&I disciplines<br>Low female enrolment in STEM and TVET<br>Male and female education imbalance<br>No equitable balance of male and female teachers<br>Out-of-school boys<br>Socio-cultural barriers that impede female participation in basic education<br>There are issues of education quality, poor education facilities<br>There are out of schoolboys<br>There is undoubtedly neglect of girls and women's education<br>Women exclusion from technological innovations<br>Women experience sexual harassment and other social vices in school |
| Gender (14) | Gender advocacy needed<br>Gender advocacy needed at federal, state, and local government levels on contraceptive usage<br>Gender education imbalance<br>Gender education imbalance<br>Gender imbalance male and female teachers<br>Gender inequalities in primary and secondary education<br>Gender mainstreaming are needed for girl's child education to promote gender mainstreaming in ST&I<br>Gender mainstreaming in all policy areas<br>Gender mainstreaming in all policy areas<br>Lack of evaluation mechanism that will effectively track and report child labour situations<br>Lack of gender mainstreaming policy and programmes<br>Lack of Gender-sensitive and programming activities<br>Lack of promotion of consistent and effective gender-responsive programs for male and female<br>There is need to mainstream women in ST&I and provide more incentives to increase women's participation in STI |
| Access (14) | Inadequate maternity and, health-care protection against women<br>Lack of access for women and girls to ST&I.Lack of access to basic education for girls.<br>Lack of access to knowledge on contraceptives<br>Limited access to finance for women<br>Only 15 percent of married women have access to contraceptive & unmet need of 36%<br>Partner opposition on the use of contraceptives,<br>Fear of side effects of contraceptives, and religious prohibitions on the use of contraceptives<br>Women and youth lack access to financial institutional support<br>Lack of mechanization serves as a disincentive to women in agriculture<br>Women lack access to agriculture inputs<br>Women lack access to finance<br>women lack access to information<br>Women lack access to land |

**Table A2.** *Cont.*

| Themes | Gendered Recognition Coded and Categorised Themes |
|---|---|
| Discrimination (13) | Discrimination Against Women<br>Discrimination against women in land holding<br>Discrimination in access to education for girls<br>Discrimination in access to education for girls<br>Discriminations in employment<br>Gender bias in land ownership<br>Marginalisation in access to resources, finance, assets, training,<br>Marginalisation in access to technology and information<br>Socio-cultural discrimination<br>There is discrimination against women workers in recruitment, remuneration, promotion, and training.<br>There is gender discrimination in land access to women<br>There is marginalisation of women in the economy<br>Women and other groups are marginalized |
| Implementation (13) | Developing proper implementation between all stake holders<br>Lack of collaboration between Government and NGOs on implementation of women's programmes<br>Lack of collaboration between implementation agencies and stakeholders on child labour issues<br>Lack of coordination for implementation of national programmes on child labour<br>Lack of framework to encourage and increase women's employment in ST&I sectors<br>Lack of legal framework that protects intended beneficiaries including children through inheritance rights, birth registration, childcare services, and breast feeding.<br>Lack of political and institutional participation on framework to promote women's ST&I.<br>Non-implementation of national teachers' policy<br>Poor enforcement of gender-based policies and institutional bias<br>There are barriers to effective implementation of child labour policy and programmes<br>There are challenges with the implementation principles in eliminating of all forms of discrimination against women<br>Weak implementation of policies to ensure inclusion of women<br>Weak legal framework to protect and promote the welfare of women and children |
| Cultural (7) | Harmful traditional practices against women<br>Socio-cultural barriers against women<br>Socio-cultural barriers impede female participation in basic education<br>Socio economic factors; economic demand, poverty, child labour, gender unfriendly educational facility<br>Distance from school and limited job opportunities and early marriage against girl-child education.<br>Socio-cultural practices against women as regards inherited lands<br>Socio-cultural, financial, or legislative encumbrances hindering women from fair participation in agriculture |

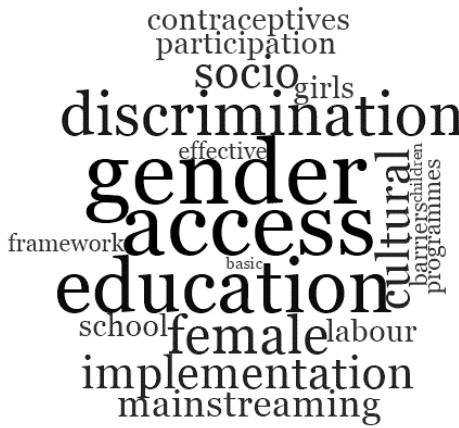

**Figure A2.** Word cloud of most frequent words in the selected policy documents.

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
