# Peer review of "How Gender Is Recognised in Economic and Education Policy Programmes and Initiatives: An Analysis of Nigerian State Policy Discourse"

_socsci, doi:10.3390/socsci10120465_

Round 1
Reviewer 1 Report
The text addresses a topic of professional, academic and institutional interest as the analysis of gender equality policies, specifically in Nigeria.
The article presents a timely academic structure. The theoretical and contextual frame of reference is considered adequate. The method describes objectives and scope. The results presented are consistent with the research objectives. The conclusions are adequately evidenced through the results presented.
Author Response
Thank you for your encouraging comments. I also see the papers as ready to be published and this has really been dealt with over the past few months. Thank you for your encouragement.
Reviewer 2 Report
The major research question this article attempts to answer is “How do public policies recognize gender in policy programmes and initiatives” in a Nigerian context. My comments are the following:
As stated in lines 35-37, little is known about how the policies and programmes benefit and work for women, and their specific recognition of women and the gender implications for women’s advancement. Therefore, in terms of achieving the policy agenda of improving the condition and welfare of women, understanding how the policies and programmes benefit and work for women is much more important than understanding how gender is recognized in the policies and programmes. Accordingly, I do think there is a lack of strong motivation statement in this article.
The research method used is “qualitative content analytical approach.” According to the authors, “an understanding of the most frequently used words within the data and then connecting the emerged themes based on their categorisation into a meaningful construction of how gender is recognized”. The term “understanding” here suggests that the six themes categorized in Appendix Table 2 is based on the author’s subjective evaluation. Therefore, Appendix Table 2 fails to provide scientific evidence to support the major conclusion reached in this article.
Author Response
|
Thank you for your encouraging comments and suggestions. I therefore made changes in the introduction section with suggestions in the aim of the study, presentation of the empirical findings and explaining the contribution of the study. In the introduction I have put forward in place the motivation for the research and the reason why gendered recognition should be studied in public policies and programmes. However, the idea of this paper is not to look at how the policies programmes benefit and work for women, but to look at how gender is recognised through public policies and programmes. But I quite agree with you on how policies might benefit and work for women, I think that has now also been added as a future research in relation to this study. The methodology was revisited, and the six main categorised themes was corrected to correspond with the only mainly appeared themes in the text to confirm the result in Appendix Table 2. |
Reviewer 3 Report
Dear authors,
The manuscript discusses how gender mainstreaming is integrated into economic and political programs, especially at the level of the Nigerian state. The paper presents in a concrete and relevant form the research context, but I consider it very important to present from the beginning the objectives of the paper, what it aims at.
Although the paper is written in plain and simple English, and the purpose of the paper is of some interest in social science practices, I must say that the paper is likely ready for publication in a local journal, as it will be of interest to students or researchers, to continue this first investigation, but fail to meet the standards of an international journal.
It is not clear how the paper contributes to the literature, nor why this context is of interest. This should be made very clear in the introduction, so that the reader is aware of what is known in relation to the questions being posed, and how this paper will add to/challenge those findings. Motivating carefully the use of this context is also important. I am interested in studies that focus on different countries (Nigeria in your case), as I feel strongly that there is a gap in the literature in terms of better understanding other institutional settings, but this needs to be carefully framed and presented, so the reader understands how the context is novel and may lead to different inferences than those obtained in the previous literature.
I consider it very important to establish from the beginning the objectives of the research, the purpose and the general question of the research, and its correlation with the obtained results.
Given the nature of the limitations, in my opinion, the paper can be published in the Social Sciences journal after the suggested alterations.
Author Response
|
This is an extremely insightful and justified remark. I have rewritten the introduction to better justify why I chose to study how gender is recognised in policy programmes and presented the aim of the study as advised. In meeting international standards, as a researcher, I must pay attention to the issue of contextualisation, as every paper needs to be contextualised in telling the story from the context of the research. We must also understand that the nature of public policies might work differently in a different contexts. As such the Nigerian public policy story might not be the same as a story from other western nations. The Nigerian context depicts an example of a developing country´s context of the way gender is integrated and treated, as this is likely to show different results from the western context.
Also, gender recognition itself is a western ideology, to be included into global public policies. Therefore, the idea of this research is to bring in western ideologies of policies in solving developing countries context of policy issues.
|
In the paper, I have added how the paper contributes to the literature and the research interest and the research aim in the introduction
Round 2
Reviewer 2 Report
Further discussion on the practical use and scholarly contribution is needed. That is, how important this study is in terms of the practical use of the findings for Nigeria? I think some of the discussion in the section “Discussion and conclusions” can be used to answer this question. As for scholarly contribution, in addition to providing complements to the extant literature, anything else is added to the existing body of knowledge through this study?
I think the author(s) need to go through the manuscript carefully since I found some sentences that are incomplete or contain grammar error, and some sentences that are not clear. The following are a couple of examples:
(Line 85-87) “Although, there has been a growing body of literature that recognises the importance and the needs for better policy discourse on the issues of gender for national development in Nigeria (Para-Mallam, 2007; (Soetan and Akanji’, 2019). -> The quoted sentence is incomplete.”
(Line 105-106) “Specifically, this article contributions are in three-folds …” -> There is an obvious grammar error in the quoted sentence.
(Line 53-55) “Scholars have reported that these policy programmes supporting entrepreneurship and women have been described to be inadequate, or failed (Bolaji 2014; Edoho, 2015; 54 Drine, & Grach, 2012) …” -> What is the term “failed” in the quoted sentence referring to?
Moreover, avoid the use of “I” in the text.
For example, in the Appendix, “After screening of the textual data in NVIVO; I further used manual …”.
Author Response
Thank you for your comments and for the insightful suggestions. I have further discussed the practical use of the study and scholarly contribution in the manuscript for complementing existing research.
I have also gone through the manuscript carefully to remove grammatical errors and added the wrong complete sentences that are incomplete. In line 85-87; 105-106; 105-106; and 53-55. I have also removed the use of “I” in the text and in the Appendix.
To summarise, I wish to thank you all for the extremely insightful comments that you have made concerning this paper. As always, sometimes, it is important to have the opinion of others in a written paper, but overall I hope I have addressed your concerns. I think all your comments and suggestions had resulted in a good substantial change in the paper which I believe have significantly improved the paper. I hope that you find the revision to meet the expectations set for a scholarly paper.
Yours sincerely,
author
Round 3
Reviewer 2 Report
Just some rephrasing on the revised manuscript as the following:
(Line 53) “But scholars have reported that these policy programmes ...” -> Change “But” to “However,”.
(Line 56) “Furthermore, these policy programmes have not also yielded much positive impact…” -> Change “have not also” to “have not yet”.
(Line 108) “This article contributions are in three-folds” -> Change “This article contributions” to “The contributions of this article”.
The three added paragraphs in lines 590-613 are suggested to move to the Introduction to highlight the practical uses and policy implications drawn from this article.
Author Response
Dear Editor,
Thank you for your comments and for the insightful suggestions. I have now further I have now made changes from removing “But” to However in line 53
I have also gone through the manuscript carefully to correct the suggestions; I have now made changes in line 56 to add “have not yet”.
Also, I have now removed some of the contributions in the introduction and merged them with the contributions from lines 590-613.
To summarise, I wish to thank you all for the extremely insightful comments that you have made concerning this paper. but overall, I hope I have addressed your concerns. I think all your comments and suggestions had resulted in a good substantial change in the paper which I believe have significantly improved the paper. I hope that you find the revision to meet the expectations set for a scholarly paper. I have also kept in mind moderate language editing. But due to the short notice to return the manuscript. I think editing can be done.
Yours sincerely,
author